# Electrically Elicited Force Response Characteristics of Forearm Extensor Muscles for Electrical Muscle Stimulation-Based Haptic Rendering

**DOI:** 10.3390/s20195669

**Published:** 2020-10-04

**Authors:** Jungeun Lee, Yeongjin Kim, Hoeryong Jung

**Affiliations:** 1Department of Mechanical Engineering, Konkuk University, Seoul 05029, Korea; jeleemanutd@konkuk.ac.kr; 2Department of Mechanical Engineering, Incheon University, Incheon 22012, Korea; Ykim@inu.ac.kr

**Keywords:** electrical muscle stimulation, haptic rendering, force response, virtual reality

## Abstract

A haptic interface based on electrical muscle stimulation (EMS) has huge potential in terms of usability and applicability compared with conventional haptic interfaces. This study analyzed the force response characteristics of forearm extensor muscles for EMS-based haptic rendering. We introduced a simplified mathematical model of the force response, which has been developed in the field of rehabilitation, and experimentally validated its feasibility for haptic applications. Two important features of the force response, namely the peak force and response time, with respect to the frequency and amplitude of the electrical stimulation were identified by investigating the experimental force response of the forearm extensor muscles. An exponential function was proposed to estimate the peak force with respect to the frequency and amplitude, and it was verified by comparing with the measured peak force. The response time characteristics were also examined with respect to the frequency and amplitude. A frequency-dependent tendency, i.e., an increase in response time with increasing frequency, was observed, whereas there was no correlation with the amplitude. The analysis of the force response characteristics with the application of the proposed force response model may help enhance the fidelity of EMS-based haptic rendering.

## 1. Introduction

Virtual reality (VR) is an immersive sensory experience simulated in a computer-generated virtual world. In VR applications, the virtual world, which refers to the contents in a virtual space, should be felt in an immersive and interactive way [1] for a user to feel the synthetic world as if it were real. Recent VR applications have provided a highly immersive visual experience, realized by advances in computer graphics technologies and immersive display devices such as head-mounted displays (HMDs) and multiple projections [2]. However, the interaction is yet to reach real-world expectations because of several technical challenges [3]. Haptics, which refers to any technology that can provide a sense of touch, is key to enabling an immersive interaction in VR [4]. Haptics provides tactile and kinesthetic perception to a user, thus enabling the sensing of the physical characteristics of virtual objects, including the mass, stiffness, and surface roughness, using specialized hardware devices [2,4,5,6]. The realism of a virtual experience can be significantly enhanced by presenting adequate haptic feedback along with high-quality visual displays [2,7].

In previous studies, numerous haptic devices have been developed to provide a high-fidelity haptic feedback in various applications including teleoperations [3,8,9,10], medical simulations [11,12,13], and virtual reality [14,15,16,17]. However, conventional haptic devices such as grounded type device, e.g., Phantom [18] and Omega [19], and exoskeleton type devices has limitations to be used in general VR applications due to the restricted workspace, complex and heavy mechanical components [15,20,21]. The haptic devices using electrical muscle stimulation (EMS) can be a feasible alternative to the conventional haptic devices by virtue of its light and flexible feature enabling the development of wearable haptic devices in the form of arm bands [14,16,22,23,24] and haptic suit [17]. EMS produces a synthetic haptic sensation by inducing muscle contractions using electrical stimulations delivered through electrodes attached to the skin surfaces [25]. The EMS-based haptic device has advantages in providing haptic sensation without requiring heavy and rigid mechanical components; however, it has several limitations in achieving a high-fidelity haptic feedback.

The lack of force response models is one of the important issues in EMS-based haptic rendering. The force response model is the mathematical representation of the relation between the electrical stimulation and the resulting force sensation. In EMS-based haptic rendering, the force response model is necessary to determine the appropriate electrical stimulation to deliver the desired force sensation to the user. However, the mechanism involved in muscle contractions induced by an electrical stimulation is complex and has not been sufficiently investigated. Moreover, it is difficult to develop the model that can be generally applied to the public because the sensitivity to electrical stimulations varies person-to-person due to the diverse physical conditions of individuals. The same stimulation may be experienced differently by different people. Most previous studies on EMS-based haptic rendering have used pre-determined stimulus patterns [14,16,23,24]. In these methods, the deliverable haptic sensation was limited to simple interactions such as recognizing certain boundaries and existence of virtual objects, and a precise haptic sensation could not be transmitted. Only few have introduced specific force models to determine the appropriate electric stimulation for delivering the desired force feedback. Kurita et al. defined a relationship between the amplitude of the electrical stimulation and the transmitted force using a sigmoid function and implemented a haptic feedback to perceive the stiffness of virtual objects using the proposed force model [23]. Kitamura et al. proposed a force response model that defines the relationship between the amplitude of the electrical stimulation and the resulting force using a first-order transfer function; however, it does not represent the nonlinear features of the force–amplitude relationship. In addition, the other parameters, such as the frequency, pulse width, and shapes, were not considered [26].

In the field of rehabilitation, EMS has been utilized to treat impaired muscle contraction caused by neuromuscular injuries. Various force response models [27,28,29,30] with sensors to measure level of muscle contraction, such as electromyography (EMG) [31,32], mechanomyography (MMG) [33,34], and piezoresistive sensors [35,36,37,38,39], have been proposed to precisely control the contraction of impaired muscles in EMS-based rehabilitation. Although these models can represent the nonlinear features of the force response with respect to the various parameters of an electrical stimulation, it remains difficult to apply these models to haptic applications because of the discrepancy of target muscles. Because most haptic sensations are perceived through the upper limb muscles, the previous models, which were developed for lower limb muscles such as the quadriceps, cannot be used without verification. In addition, the complexity of the previous models makes it difficult to use them for haptic rendering. The previous models representing the force–amplitude relationship [28] and force–pulse width relationship [30] include 9 to 10 adjustable coefficients. Too many variables in the model can not only lead to an overfitting problem, but also decrease the computational efficiency, which is critical in real-time haptic applications.

This paper introduces a force response model that can represent the transient response of an electrically elicited muscle contraction force, and experimentally validated the feasibility of the model for haptic applications. The force response characteristics of the forearm muscles, including the peak force and response time, with respect to the amplitude and frequency of the electrical stimulation were investigated through the experiment involving ten healthy subjects. The experimental results were analyzed quantitatively using the proposed force response model. The main contributions of the paper are as follows.
The simplified force response model developed for lower limb muscles in the field of rehabilitation was introduced for EMS-based haptic rendering, and its feasibility for the forearm extensor muscles was experimentally verified. Thus, the model can be utilized for EMS-based haptic rendering given its feasibility for upper limb muscles.The force response characteristics, including the peak force and response time, were identified through the experiment based on a quantitative evaluation using the force response model. The peak force–amplitude relationship was assumed as an exponential function and verified by comparing the estimated peak force with the experimentally measured peak force. The response time characteristics were also identified by performing a quantitative evaluation using the force response model fitted to the experimental data. The presented results are expected to contribute to our understanding of the transient and steady-state features of the muscle contraction force response.

## 2. Materials and Methods

### 2.1. Principles of EMS-Based Haptic Rendering

In EMS-based haptic rendering, a haptic sensation is delivered through muscle contraction elicited by an electrical stimulation. Figure 1 shows the concept of EMS-based haptic rendering. In the real world, a human perceives a kinesthetic sensation through a reaction force at the point of contact when he/she physically touches an object. However, in case of VR, the user cannot experience any reaction force since there is no physical contact between the user’s hand and the virtual object. In EMS-based haptic rendering, the reaction force is delivered by obstructing the user’s intended motion by contracting specific muscles, corresponding to a motion opposite to the user’s intention, using adequate electrical stimulation. When the user presses the virtual object, as shown in Figure 1b, the extensor muscles in the forearm are contracted to block the user’s hand passing through the surface of the virtual object and deliver a corresponding haptic sensation. In this context, it is important to identify the characteristics of muscle contraction under the induced electrical stimulation. First, the saturated peak force of the muscle contraction with respect to the EMS parameters should be identified to present the desired haptic sensation transparently. It is necessary to determine the values of the EMS parameters to transmit the desired haptic sensation to the user. Second, the response time of the muscle contraction force to the electrical stimulation should be identified. Because the muscle contraction force gradually increases over a certain time interval to reach the target force, it is necessary to determine the response time of the force to deliver a haptic sensation at the exact timing. In this paper, these two features of the electrically elicited muscle contraction were investigated based on a mathematical model of the muscle contraction force response.

### 2.2. Simplified Force Response Model of Muscle Contraction Elicited by Electrical Stimulation

This section presents the force response model of the muscle contraction elicited by the electrical stimulation. Based on the previous force model proposed by Ding et al. [27], we present the derivation of a simplified force response model incorporating a biphasic pulse input with amplitude modulation using an exponential function for EMS-based haptic rendering. The presented force response model was used not only to simulate the force response with respect to the given EMS parameter values but also to quantitatively evaluate the characteristics of muscle contraction in the experiment.

#### 2.2.1. EMS Parameters

The force response model representing the relationship between the given electrical stimulation and the elicited muscle contraction force can be simply expressed as a function of the electrical stimulation parameters as follows.
(1)$$F\left(t\right)=f\left(P_{WF},\;P_{PW},\;P_{FQ},\;P_{AM}\right)$$
where $P_{WF},\;P_{PW},\;P_{FQ},\;\mathrm{and}\;P_{AM}$ are the four parameters of the EMS representing the waveform, frequency, pulse width, and amplitude of the electrical stimulation given by a pulse train, respectively. Figure 2 shows the shape of the pulses with respect to these parameters. Each parameter contributes to the muscle contraction force in a different manner, and this should be carefully determined not only to produce the desired muscle contraction force but also to minimize the negative effects of the EMS, such as muscle fatigue, pain, and discomfort, to provide an immersive haptic sensation. The important features of the EMS parameters that should be considered for immersive haptic rendering are as follows.

*Waveform* ($P_{WF}$): The type of waveform used affects a user’s sensation to the electric signal, and it should be determined to minimize discomfort. There are two types of waveforms used in EMS applications which are classified based on the polarity: monophasic and biphasic rectangular waveforms. Monophasic waves have a fixed polarity and deliver a unidirectional electric stimulation. Biphasic waves contain positive and negative pulses that deliver a bidirectional electric stimulation. Studies have shown that biphasic waves are more advantageous considering muscle fatigue [40], comfort [41], and skin damage [42]. Thus, we chose a pulse train with a biphasic rectangular waveform for deriving the force response model and for experimental validation.

*Pulse width* ($P_{PW}$): The pulse width is the elapsed time between the rising and falling edges of a single pulse, and it affects the magnitude of the muscle contraction force. An increase in the pulse width increases the magnitude of the force generated. The muscle contraction force gradually increases as the pulse width increases up to 400 µs [43], which is the upper limit of the pulse width to prevent pain and muscle fatigue. Studies have shown that the effect of pulse width on the muscle contraction force is not significant compared with those of the frequency and amplitude. Thus, this study assumed the pulse width as a static parameter and used the maximum pulse width (400 µs) in the experimental study.

*Frequency* ($P_{FQ}$): The frequency is the number of pulses per second, and it affects the magnitude of the muscle contraction force [42,44,45]. A high-frequency signal can result in a higher muscle contraction force; however, a high-frequency electrical stimulation typically causes pain and discomfort [46]. Both high-frequency (>50 Hz) [42,47] and low-frequency (<20 Hz) [48,49] signals can cause muscle fatigue; thus, such frequency ranges should be avoided. In the experiment, the frequency was set in the range of 20~50 Hz.

*Amplitude* ($P_{AM}$): The amplitude is the intensity of the electrical current, i.e., the magnitude of the pulse, as shown in Figure 2. High-amplitude signals induce a greater muscle contraction force; however, they should be carefully determined because a too intensive stimulation can cause pain and muscle fatigue. Because the impedance between the skin and the muscle fibers varies with the physical conditions, such as the fat mass composition, an identical amplitude of the electrical stimulation can be experienced in a different manner [50]. This study determined the range of amplitudes under a threshold that causes pain and muscle fatigue for each subject and applied these amplitudes in the experiment.

#### 2.2.2. Force Response Model for Biphasic Pulse Input

The muscle force response to an external electrical stimulation can be modeled using two first-order differential equations as follows [27,51].
(2)$$\frac{dC}{dt}=u\left(t\right)-\frac{C}{\tau_{c}}$$
(3)$$\frac{dF}{dt}=A\frac{C}{C+K}-\frac{F}{\tau_{1}+\tau_{2}\frac{C}{C+K}}$$
where C and F are two state variables representing the amount of Ca^2+^–troponin complexes and the muscle contraction force, respectively, u is the external input of the electrical stimulation, and the five constant parameters $A,\;K,\;\tau_{c},\tau_{1},\;\mathrm{and}\;\tau_{2}$ represent the force scaling factor, sensitivity of strongly bound crossbridges to C, time constant controlling the rise and decay of C, time constant of force decline in the absence of strongly bound crossbridges, and time constant of force decline in the presence of strongly bound crossbridges, respectively [52]. Equation (2) demonstrates the biochemical procedure of the concentration of Ca^2+^–troponin complex with respect to the external electrical stimulation. Equation (3) represents the biophysical procedure for developing muscle contraction forces with respect to the calcium concentration derived from a linear spring, damper, and motor connected in series [51]. The external input u(t) given by a series of pulse trains can be represented by the summation of the impulse functions, as follows [29].
(4)$$u\left(t\right)=\overset{n}{\underset{i=1}{\sum }}\delta \left(t-t_{i}\right)$$
where $t_{i}$ is the time of pulse excitation. In this study, we selected pulses with a biphasic waveform to minimize discomfort during the electrical stimulation. In the literature, it has been identified that there was no significant difference in force generation [40] as well as concentration of Ca^2+^-troponin complex [46] between monophasic and biphasic waveforms. Thus, monophasic and biphasic waveform can be assumed to have same effect to the force generation, and Equation (4) can be used for the biphasic waveform of external input u(t).

#### 2.2.3. Amplitude Modulation

The force scale factor A in Equation (3) can be determined by the peak force for a given amplitude of the pulse [30]. In previous studies, the peak force was measured over a wide range of pulse amplitudes for the lower limb muscles, and the peak force–amplitude relationships could be represented using a sigmoid function, as shown in Figure 3 [23,53,54,55]. Although the peak force varies with the pulse amplitude between a motor threshold ($I_{MT}$) and a saturation threshold ($I_{ST}$), which represent the minimum pulse amplitudes for triggering and saturating the muscle contraction force, respectively, the full range of the pulse amplitude need not be considered for determining A in EMS-based haptic rendering. Because an excessive electrical stimulation with a high-amplitude pulse can cause discomfort [45] and pain [45], the available range of the amplitude should be limited carefully considering the discomfort and safety of the user. The range of pulse amplitudes in EMS-based haptic rendering can be determined by the other threshold values: pain threshold ($I_{PT}$). The pain threshold was defined as the minimum amplitude that causes pain to the user. Studies have shown that a user begins to feel discomfort and pain when the amplitude of the electric stimulation exceeds three times the motor threshold [56]. In this study, the value of $I_{PT}$ is determined conservatively as $I_{MT}\times 1.5$ by applying safety factor of 2.0, and the available range of the amplitude in EMS-based haptic rendering is set with $I_{MT}$ and $I_{PT}$. Figure 3 shows the available range of the amplitude in EMS-based haptic rendering, denoted as the haptic range. Because the haptic range is included in the acceleration region of the sigmoid function, the peak force–amplitude relationship can be simply represented by an exponential function, as follows.
(5)$$F_{PK}\left(I\right)=ae^{bI}$$
where I is the excitation amplitude of the pulse, and a and b represent the coefficients of the exponential function. The force scaling factor A in Equation (3) can be replaced by $F_{PK}$, as follows.
(6)$$A\left(I\right)=\frac{A\left(I_{0}\right)}{F_{PK}\left(I_{0}\right)}F_{PK}\left(I\right)$$
where $I_{0}$ represents the specific amplitude value in which the force scaling factor is initially evaluated. With Equation (6), once the force scaling factor A is evaluated at a specific amplitude, it can then be estimated in the other amplitude ranges using the value of $F_{PK}\left(I\right)$.

### 2.3. Experimental Evaluation of the Force Response

The force response model was experimentally validated for the forearm extensor muscles which can be used for presenting the haptic sensation related to the wrist flexion motion frequently occurring in various VR applications. Two important features of the force response for EMS-based haptic rendering, including the peak force and response time, with respect to the amplitude and frequency of the electrical stimulation were also identified in the experiment.

#### 2.3.1. Participants

Ten healthy subjects, nine male and one female (age 26.4 ± 1.96 years; height 174.3 ± 8.25 cm; weight 76.6 ± 14.9 kg), participated in the experiment. The experimental protocol was approved by the Konkuk University Institutional Review Board (7001355-201909-HR-337). All the subjects were instructed not to exercise excessively the day before the experiment. In addition, the purpose, procedure, and risks of the experiment were explained in detail.

#### 2.3.2. Experimental Setup

Figure 4 shows the experimental setup including an EMS device, a torque sensor, a DAQ board, an anchoring jig, and a laptop. A portable EMS device (Model rehamove3, Hasomed, Germany) was used to produce the desired electrical signals and send to the target muscles. Four extensor muscles of the forearm including extensor digitorum, extensor carpi radialis longus, extensor carpi ulnaris, and extensor digiti minimi were set as the target muscles. Two EMS electrodes (5 × 5 cm, ValuTrode, Denmark) were attached to cover the motor points of these extensor muscles. Initially, the subjects were instructed to sit on a chair and place the dominant arm on the anchoring jig. The subjects’ arm and wrist were tightly fixed on the jig with straps to constrain the motion of the forearm and allow only one degree-of-freedom motion of the wrist in the transverse axis. The tightness of the strap was carefully adjusted not to disturb the contraction of the forearm muscles by testing the muscle contraction with test electrical signals. The hand was fixed to an L-shaped fixture connected to the torque sensor using the strap. The force of the wrist extension due to the contraction of the four extensor muscles of the forearm was measured by the torque sensor based on the force–torque relationship F=τ/L, where F represents the force exerted onto the L-shaped fixture at the point connecting the subject’s hand, and L is the vertical distance from the axis of the torque sensor to the point at which the force is exerted. τ represents the torque measured by the torque sensor (NT-200KC, Sensor solution, Korea). τ measured at a sampling rate of 1 kHz was amplified using an amplifier (ST-AM100, Sensor solution, Korea), sent to the laptop through the DAQ board (NI USB 6008, National Instruments, USA), and converted to the force exerted by the wrist extension. Because the motion of a subject’s hand is tightly constrained to wrist extension, we assumed that the force measured by the sensor is equivalent to the force resulting from co-contraction of the four extensor muscles of the forearm. The collected force data were processed using a moving average filter whose window size and cutoff frequency were 40 ms and 11.1 Hz respectively to remove high-frequency noises. After completing the experimental setup, the position of the electrodes was adjusted by testing the muscle contraction with the test signal whose frequency, pulse width, and amplitude were set to 20 Hz, 400 μs, and above 8 mA, respectively. The positions of the electrodes were adjusted gradually until a clear contraction of the target muscles was confirmed.

#### 2.3.3. Experimental Procedures

Four EMS parameters ($P_{WF},\;P_{PW},\;P_{FQ},\;\mathrm{and}\;P_{AM}$) were considered in the experiment, as listed in Table 1. $P_{WF}$ and $P_{PW}$, which are denoted as static parameters, were set to their pre-defined static values and were not varied during the experiment. Considering muscle fatigue and pain, $P_{WF}$ and $P_{PW}$ were set to rectangular biphasic pulse and 400 μs, respectively, which are known to minimize muscle fatigue and pain [41,46,47]. $P_{FQ}$ and $P_{AM}$, which are denoted as control parameters, were varied to investigate the characteristics of the force response with respect to the values of these control parameters. Three frequency values (20, 30, and 40 Hz) were used in the experiment to identify the peak force and respond time characteristics with respect to the frequency. The available frequency band (20–40 Hz) was determined as the band to avoid muscle fatigue and discomfort. Five amplitude values, denoted by $I_{1},\;I_{2}\cdots I_{5}$, were used in the experiment. The amplitude values were determined not to exceed the pain threshold $I_{PT}$, shown in Figure 3. For each subject, the experiment proceeds with four steps as follows.

*STEP 1 (Posture set up):* At the beginning of the experiment, the subject’s right arm is fixed on the experimental set up as presented in previous section.

*STEP 2 (Electrode attachment):* After fixing the subject’s right arm, one directed the subject to do wrist extension, and find the point that shows the greatest contraction by visual inspection. Two square electrodes with a side length of 5 cm attached at the designated point with intervals of 6–7 cm to cover four motor points of the wrist extensor muscles as shown in Figure 5. And then the position of the electrodes slightly adjusted by testing the muscle contraction using test electrical simulation (20 Hz, 400 μs, 8 mA) until the clear muscle contraction is confirmed.

*STEP 3 (Determination of*$I_{MT}$): The EMS device begins to stimulate the subject’s target muscle by gradually increasing the amplitude from 0 mA with an interval of 0.5 mA to identify the motor threshold $I_{MT}$. Once the wrist extension is detected, the corresponding amplitude value was designated as the motor threshold. If the force measured by the torque sensor exceeds the threshold value (0.015 N), we assumed wrist extension is detected. In this procedure, the waveform, pulse width, and frequency of the electrical signal were set to rectangular biphasic, 400 μs, and 30 Hz, respectively. Since the variation of $I_{MT}$ according to the frequency was less than 0.5 mA, the $I_{MT}$ measured with 30 Hz signal was used as representative motor threshold value of each subject. And then, $I_{1}$ was set to $I_{MT}$, and the remaining four amplitude values ($I_{2}$~$I_{5}$) were set by increasing the value of $I_{1}$ with a uniform interval so that the maximum amplitude $I_{5}$ is equal to the pain threshold $I_{PT}$.

*STEP 4 (Force response acquisition for 15 cases of electrical stimulation):* After determining stimulation amplitudes ($I_{1}$~$I_{5}$) for the subject, the EMS devise begins to send a pulse train with variable amplitudes and frequencies, as shown in Figure 6. The pulse train consists of three stimuli and three breaks with different frequencies. One second stimulus followed by a nine second break was repeated three times in one pulse train by changing the frequency of each stimulus in an ascending order (20, 30, and 40 Hz). The break, which continues for nine seconds between the stimuli, was assumed to be sufficient for the contracted muscles to be relaxed. Five pulse trains were sequentially transmitted to the target muscle by changing the amplitude of the pulse train in an ascending order ($I_{1}$, $I_{2}$, $I_{3}$, $I_{4}$, and $I_{5}$). The force responses for 15 cases of different electrical stimulation were measured for each subject.

### 2.4. Parameter Identification

The muscle contraction force profiles of each subject acquired by the sensor at a sampling rate of 1 kHz were used to estimate the parameters of the force response model. First, the force profile acquired for fifteen consecutive stimuli were segmented manually to fifteen force responses corresponding to a single stimulus. For each segmented force response, the parameters of the mathematical model were determined as the values that minimize the errors between the experimental and simulated force responses of the mathematical model through an iterative optimization procedure. The simulated force response was produced by numerically integrating Equations (2) and (3) with the estimated parameter values. The fourth-order Runge–Kutta method with a fixed time interval of 0.2 ms was used for the numerical integration. The initial values of the state variables (C and F) in the equations were set to zero.

Two objective functions were used to determine the optimal parameters of the force response model. The five parameters ($A,K,\tau_{c},\tau_{1},\;\mathrm{and}\;\tau_{2}$) of the force response model expressed in Equations (2) and (3) were obtained by minimizing the following objective function [28,30]
(7)$$C_{1}\left(A,K,\tau_{c},\tau_{1},\tau_{2}\right)=\sum_{i=1}^{n}{\left(F^{pred}\left(t_{i};A,K,\tau_{c},\tau_{1},\tau_{2}\right)-F^{exp}\left(t_{i}\right)\right)}^{2}$$
where n is the number of samples in the force response, and $F^{pred}\left(t_{i};\tau_{c},A,K,\tau_{1},\tau_{2}\right)$ and $F^{exp}\left(t_{i}\right)$ represent the simulated force response with the estimated parameter values and the measured force in the experiment at the sampling time $t_{i}$, respectively. The trust-region-reflective algorithm was used to determine the optimal parameter values [29]. To avoid the local minima, the initial values of the parameters were set randomly in a predefined range, and the iterative optimization procedure was continued until the value of the objective function was less than $10^{-6}$. The parameters of $F_{PK}$ (a, b) were obtained by minimizing the following objective function [28,30].
(8)$$C_{2}\left(a,b\right)=\sum_{k=1}^{m}{\left(F_{PK}^{pred}\left(k;a,b\right)-F_{PK}^{exp}\left(k\right)\right)}^{2}$$
where *m* is the number of the peak force samples acquired for each frequency, $F_{PK}^{pred}\left(k;a,b\right)$ and $F_{PK}^{exp}\left(k\right)$ represent the estimated and measured peak force, respectively, and k is the index of the sample. The parameter estimation of the peak force was performed separately for the samples obtained in response to the three different stimulation frequencies.

## 3. Results

The force response characteristics of the forearm extensor muscle were identified using the experimental data. First, the feasibility of the exponential estimation of the peak force was evaluated by comparing the estimated peak force value with the measured value. The root-mean-square-error (RMSE), normalized RMSE (NRMSE) and the coefficient of determination ($R^{2}$) were used for the accuracy evaluation. NRMSE was calculated by dividing the RMSE by the peak force to represent the percentage error to the peak force. Second, the reliability of the force response model was evaluated by comparing the simulated force response, produced by the mathematical model using the estimated parameters, with the experimental force response. Third, the response time of the muscle contraction force was identified.

### 3.1. Peak Force–Amplitude Relationship

Table 2 lists the peak force values for all the subjects, extracted from the force responses measured in the experiment with respect to the stimulation frequencies. The values in the parenthesis represent the corresponding amplitude values. The minimum and maximum values of the peak force presented for each frequency represent the peak force values measured with the minimum amplitude ($I_{1}$) and maximum amplitude ($I_{5}$), respectively. The results show that the peak force varies in a wide range from 1.47 N to 8.57 N at the minimum amplitude ($I_{1}$) and frequency (20 Hz) and from 7.66 N to 28.0 N at the maximum amplitude ($I_{5}$) and frequency (40 Hz). Moreover, identical tendencies, i.e., an increase in the peak force with increasing frequency, were observed for all the subjects. The lowest peak force was observed for the subject K01, and the corresponding peak force values were 1.47, 1.68, and 2.16 N at frequencies of 20, 30, and 40 Hz, respectively. Moreover, the highest peak force was observed for the subject K03, and the corresponding peak force values were 23.7, 26.7, and 28.0 N at the three frequencies, respectively.

Table 3 lists the parameters of the exponential estimation, RMSE, NRMSE and $R^{2}$. The values presented for each frequency are the averaged values for all the subjects. As listed, the value of $R^{2}$ is higher than 0.96 at all the frequencies, indicating that the exponential estimation of the peak force is in good agreement with the peak force of the muscle contraction. RMSE values are also less than 10% of the peak force for all frequencies. The parameter a increases proportionally with the frequency in the range of 0.497~0.734 N, whereas the parameter b remains largely the same at all the frequencies. Figure 7 shows the exponential estimation of the peak force (PK) with respect to the amplitudes with the measured peak force samples for the cases presenting best (K05) and worst (K03) estimation results. The $R^{2}$ values corresponding to the best estimation are 0.98 (20 Hz), 0.99 (30 Hz), and 0.97 (40 Hz), whereas the $R^{2}$ values corresponding to the worst case are 0.9 (20 Hz), 0.93 (30 Hz), and 0.9 (40 Hz). The graphs clearly show an exponential relationship between the peak force and the amplitude with the tendency of the peak force, which increases with the stimulation frequency not only in the best case but also in the worst case. In Figure 7, the graphs shown on the right present the peak force estimation normalized by the maximum value of peak force ($F_{PK}\left(I_{5}\right))$ at each frequency. The normalized curves are largely identical regardless of the frequency.

Figure 8 shows the correlation between the estimated and measured peak force for peak force samples at all combinations of pulse amplitude and frequencies. The slope of the regression, $R^{2}$, and RSME value were 0.92, 0.97, and 0.85 N respectively.

### 3.2. Accuracy of the Force Response Model

Table 4 lists the resulting parameters of the model for all subjects with the corresponding RMSE, NRMSE and $R^{2}$ values. The parameter values were estimated using the force response measured at the maximum amplitude $I_{5}$ [28]. The averaged $R^{2}$ of the estimation was 0.93, indicating that the force response model provides reliable estimation results similar to the results presented in literature [27]. The averaged NRMSE was 8.06% which means averaged RMSE was less than 10% of the peak force. The most accurate prediction of the force response was observed in the result of the subject K01 that showed 0.97, 0.64 N, 5.97% of $R^{2}$ RSME, and NRMSE, respectively. In the worst case (K05), $R^{2}$, RSME, and NRMSE were 0.89, 1.47 N, and 10.3%, respectively. Figure 9 show the estimated and measured force response of the subject K10. The graphs demonstrated that the simulated force response closely fitted to the measured force response. In these graphs, the oscillations vanished in the experimental force response because those were filtered out by the moving average filter.

Table 5 and Figure 10 shows the accuracy of the force response model for all amplitude and frequency values. In this analysis, the parameters of the model were evaluated using the force response measured at the maximum amplitude ($I_{5}$), and tested with the force response measured at the other amplitudes. This analysis was conducted to identify that the force response model evaluated on the specific frequency and amplitude is valid for other frequency and amplitude values. In the literature, it has been reported that the parameter estimation using the force response acquired with high amplitude electrical stimulation provides superior estimation results [37,39]. Only the force scaling factor A was adjusted for each amplitude using Equation (6), whereas the other parameter values were identically applied with the values listed in Table 4. The averaged values of $R^{2}$ were 0.82, 0.88, 0.86, and 0.88 for $I_{1}$, $I_{2}$, $I_{3}$, and $I_{4}$, respectively that are slightly lower than the $R^{2}$ value (0.93) evaluated on $I_{5}$. The averaged RMSE values were lies within the range from 10.8% to 13.1% of the peak force for $I_{1}\sim I_{4}$ that are slightly larger than that of $I_{5}$. This result shows that the force response model evaluated in specific frequency and amplitude can be used to estimate the force in the other frequencies and amplitudes within acceptable error bounds.

### 3.3. Characteristics of the Response Time

The response time of the muscle contraction force was evaluated using the time interval required to reach the peak force, as follows.
(9)$$T_{RT}=T_{0.9}-T_{0.1}$$
where $T_{RT}$ is the response time, and $T_{0.1}$ and $T_{0.9}$ represent the times at which the force reaches 10% and 90% of the peak force, respectively. The reference time (t=0) to determine $T_{0.1}$ and $T_{0.9}$ was defined as the moment that the force response exceeds the threshold (0.015 N). Table 6 lists the averaged response time of the force profile measured in the experiment. The time values $T_{0.1}$, $T_{0.5}$, and $T_{0.9}$ were calculated by averaging the values obtained from 15 cases of the electrical stimulation (three frequencies and five amplitudes), and the response time $T_{RT}$ was calculated using averaged $T_{0.1}$ and $T_{0.9}$ with Equation (9). The force reached 10% of the force within 15 ms and 50% of the peak force within 113 ms for all the subjects. The response time $T_{RT}$ varies in the range of 184–349 ms for each subject, and the averaged $T_{RT}$ is 283.5 ms with a standard deviation of 61.3 ms. Table 7 lists the response time in terms of the frequency. As the frequency increases from 20 to 40 Hz, the averaged response time increases from 261.4 ms to 301.3 ms, whereas the slope of the response time gradually decreases as shown in Figure 11a. Unlike the frequency, there is no change in the response time with respect to the pulse amplitude as shown in Figure 11b.

## 4. Discussion

The aim of this study was to identify the force response characteristics of the muscle contraction elicited by an electrical stimulation for EMS-based haptic rendering. Two important features of the force response, including the peak force and response time, were investigated experimentally using a simplified mathematical model of the force response. This section presents the implications and insights obtained through the experimental analysis of the force response with respect to the frequency and amplitude of the electrical stimulation in the context of haptic rendering applications. The contents presented in this section are pertinent to our understanding of the important features of the muscle contraction force response and thus can help improve the accuracy and reliability of EMS-based haptic rendering.

### 4.1. Peak Force–Amplitude Relationship

The range of the pulse amplitude applied in the experiment ($I_{1}\sim I_{5}$) was determined by the motor threshold $I_{MT}$ measured for each subject, and it varied for the subjects in the range of 7~12 mA. Moreover, the peak force measured at $I_{MT}$ varied in a relatively wide range of 1.47~8.57 N. This result shows that the absolute value of the peak force corresponding to the given stimulation amplitude is difficult to be determined uniquely because the elicited force response varies with an individual’s physical condition even under an identical stimulation. Nevertheless, the tendency of the peak force variation with respect to the frequency was clearly observed in the experimental data. The peak force–amplitude relationship could be represented by an exponential function, and the accuracy of the model was confirmed using the NRSME (<8.35%) and $R^{2}$(>0.96) values as presented in Table 3. Regarding the two parameters of the exponential function (a, b), a frequency-dependent tendency was identified. The parameter a, which determines the scale or magnitude of the estimation, increased with increasing frequency, leading to a higher peak force in the high-frequency stimulation. However, the parameter b, which determines the shape of the estimated result, did not vary with respect to the frequency, and this feature resulted in a similar shape of the normalized exponential estimation at different frequencies. This result implies that the peak force can be estimated using a unique exponential function determined at a specific frequency and it can be applied to estimate the peak force in the other frequency range by introducing a frequency-dependent scaling factor.

### 4.2. Accuracy of the Force Response Model

This study introduced a simplified mathematical model of the force response and validated its reliability for haptic applications based on the experimental force profiles acquired from the forearm extensor muscles. Although a few validation studies have been conducted on force response models, it was necessary to evaluate whether the model is valid for forearm extensor muscles because the previous validations were conducted only for lower limb muscles. This study confirmed that the simplified mathematical model of the force response is valid for forearm extensor muscles, which are mainly involved in EMS-based haptic rendering. To the best of our knowledge, this study is the first to introduce a mathematical model of the force response for EMS-based haptic rendering. The parameters of the force response model were identified for each subject using the measured force response data at the maximum amplitude ($I_{5}$), and the accuracy of the model was evaluated by comparing the simulated force response with the measured force response not only at $I_{5}$ but also at the other amplitudes ($I_{1}\sim I_{4})$. When evaluating the accuracy at the amplitudes $I_{1}\sim I_{4}$, we adjusted only the force scaling factor A based on the peak force–amplitude relationship. The validation presented highly accurate estimation results (NRMSE = 0.93, $R^{2}$ = 8.06) in the force response of $I_{5}$. In the case of the force response measured at the other amplitudes ($I_{1}\sim I_{4}$), the accuracy results were slightly lower than that at $I_{5}$; nevertheless, the model provided reliable estimation results (NRSME < 13.1%, $R^{2}$ > 0.82) for haptics application. Considering the just noticeable difference (JND) of the force (10~15%) [57], the accuracy results were in the acceptable range for haptics applications. The validation results demonstrated the feasibility of introducing a force response model for EMS-based haptic rendering.

### 4.3. Response Time

Identifying the features of the response time is important to deliver the desired force feedback at the exact timing in real-time haptic applications. In this study, the response time of the force was evaluated at various frequencies and amplitudes using the mathematical model fitted to the experimental data. The slope of the force response decreases as it reaches to the peak force that results as shown in the shape of the force responses. To evaluate the response time, $T_{0.1}$, $T_{0.5},$
$T_{0.9}$, and $T_{RT}$ were calculated for all the force responses. As expected from the shape, $T_{0.9}-T_{0.5}$ was greater than $T_{0.5}-T_{0.1}$ for all subjects, as presented in Table 6. From the response time analysis, it was confirmed the response time $T_{RT}$ varies with the frequency: $T_{RT}$ increased with increasing frequency. The force response in the case of a high-frequency electrical stimulation has a longer response time than that in the case of a low-frequency electrical stimulation. The response time can also be expected by the model for muscle contraction. The term related to the response time in the force response is $\tau_{1}+\tau_{2}\frac{C}{C+K}$ in Equation (3), and an increase in $\tau_{1}+\tau_{2}\frac{C}{C+K}$ causes a large response time. As the frequency increases, the value of C increases which resulted in the large response time. However, the pulse amplitude only affects the force scaling parameter A, not C, thus $\tau_{1}+\tau_{2}\frac{C}{C+K}$ is constant with respect to the change in the pulse amplitude [58].

### 4.4. Limitation

The proposed model can be used to estimate the muscle force response according to the frequency and amplitude. However, there remains a limitation that should be resolved in the future research. The force response characteristics can be influenced by the posture, but this study only considered the force response of isometric muscle contraction in a fixed posture. Correlation between the force response and the posture should be investigated to be applicable to more diverse situations in the future research. In addition, the force response characteristics presented in this paper does not represent the force response of actual muscle contraction, but overall force response of wrist extension caused by co-contraction of forearm extensor muscles. For precise control of each muscle contraction, further research is required to measure the level of individual muscle contraction using sensors such as piezoresistive sensors.

## 5. Conclusions

In this study, we analyzed the force response characteristics of forearm extensor muscles for EMS-based haptic rendering by introducing a simplified force response model with amplitude modulation using an exponential function. The proposed force response model can be utilized to predict not only the transient behavior but also the steady-state characteristics of the force response for determining appropriate EMS parameters that can provide the desired haptic sensation. The features of the PF and RT with respect to the frequency and amplitude were identified using the experimental force response data, and this result can be utilized to implement EMS-based haptic rendering in various haptic applications.

## Figures and Tables

**Figure 1 sensors-20-05669-f001:**
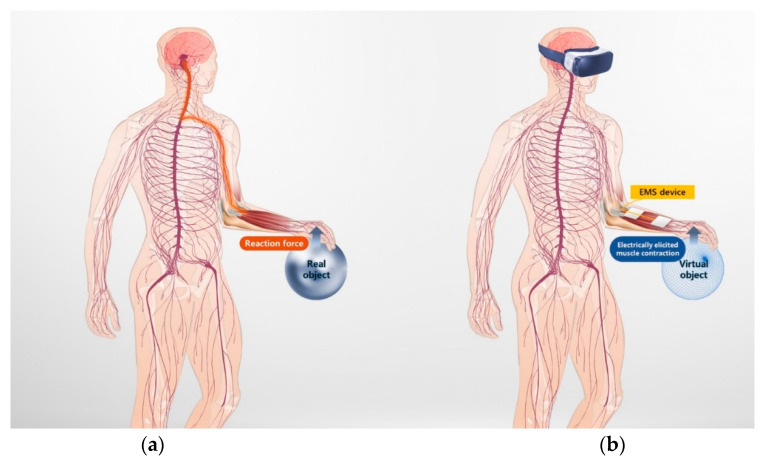
Representation of perceiving kinesthetic sensation when pressing a sphere: (**a**) when user presses the sphere in the real world; (**b**) when user presses a virtual sphere in a virtual world, where the EMS induces a virtual reaction force.

**Figure 2 sensors-20-05669-f002:**
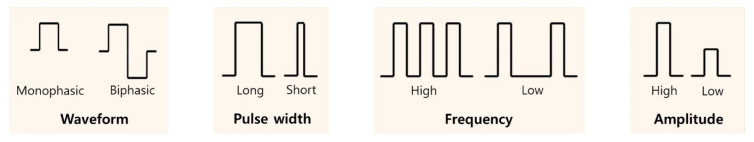
Shape of pulse with respect to the EMS parameters.

**Figure 3 sensors-20-05669-f003:**
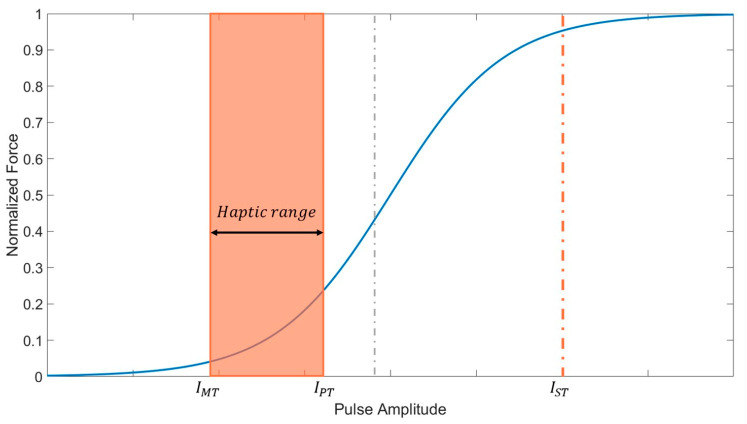
Representation of the peak force curve with respect to the pulse amplitude. The orange box represents available range of the amplitude in EMS-based haptic rendering.

**Figure 4 sensors-20-05669-f004:**
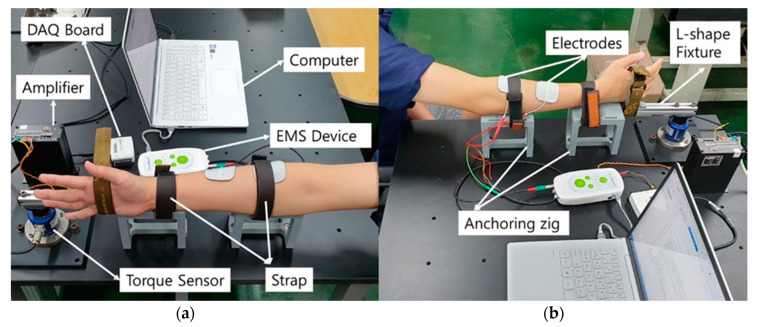
Representation of the experimental setup: (**a**) left view (**b**) right views of the experiment configuration.

**Figure 5 sensors-20-05669-f005:**
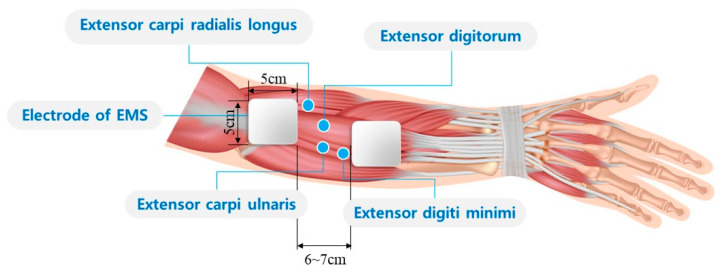
Representation of electrode attachment with motor points of the four extensor muscles.

**Figure 6 sensors-20-05669-f006:**
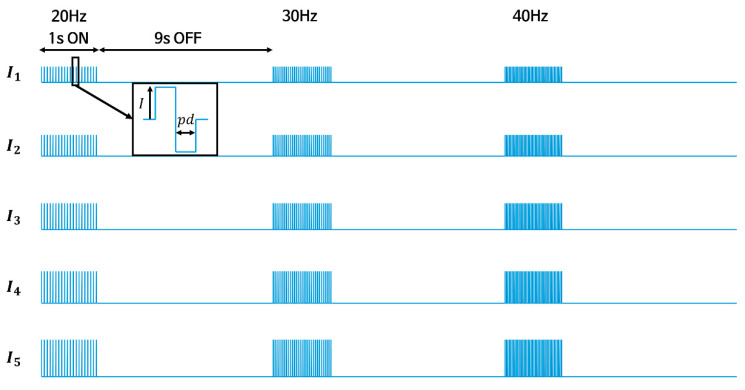
The pulse trains used in the experiment for one subject. The biphasic pulse trains of three frequencies and five amplitudes were sequentially transmitted with nine second break.

**Figure 7 sensors-20-05669-f007:**
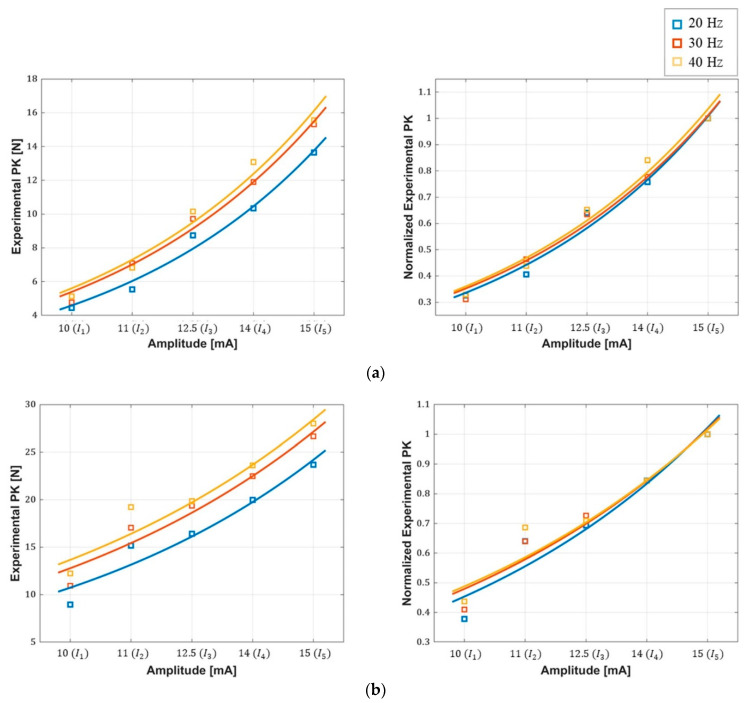
Plots of exponential estimation of the peak force: (**a**) best estimation (**b**) worst estimation.

**Figure 8 sensors-20-05669-f008:**
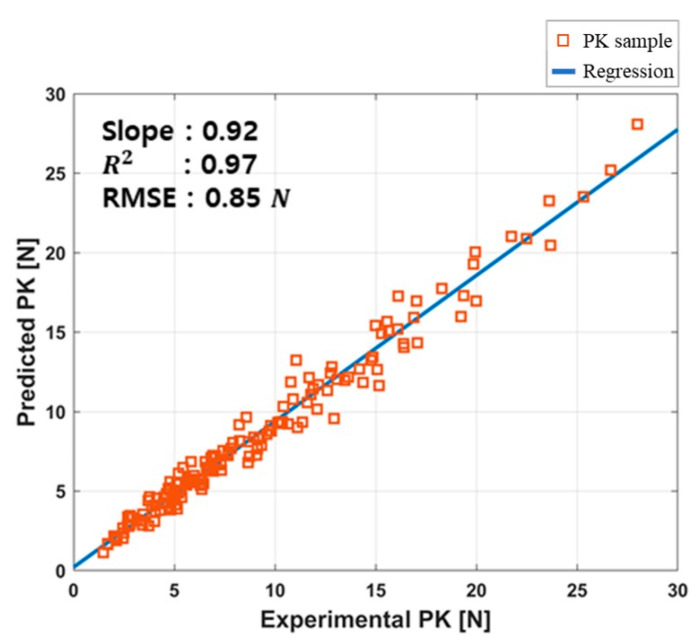
Correlation plots of estimated and measured peak force.

**Figure 9 sensors-20-05669-f009:**
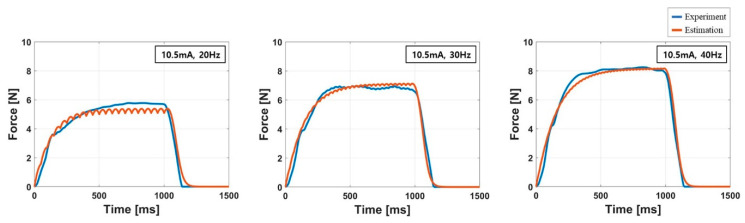
Simulated and measured force responses of the subject K10 at the maximum amplitude $I_{5}$. The red and blue lines represent the estimated and experimental force response, respectively.

**Figure 10 sensors-20-05669-f010:**
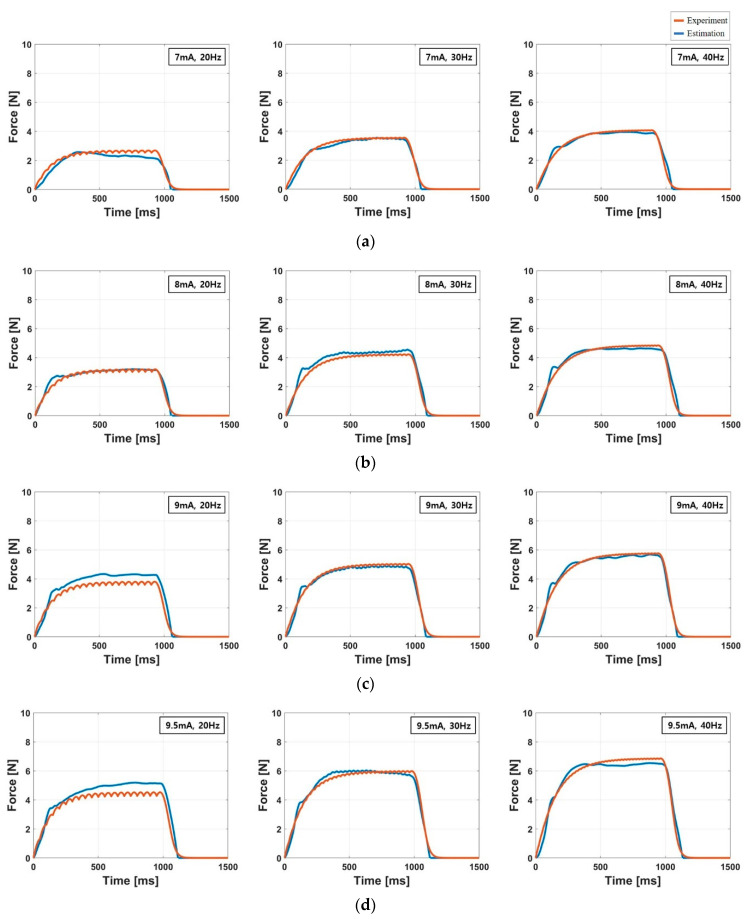
Simulated and measured force responses of the subject K10 at the amplitude $I_{1}\sim I_{4}$. The parameter values used for the simulated force response were determined by the force response measured at $I_{5}$. (**a**) force response at $I_{1}$, (**b**) force response at $I_{2}$, (**c**) force response at $I_{3}$, (**d**) force response at $I_{4}$. The red and blue lines represent the estimated and experimental force response, respectively.

**Figure 11 sensors-20-05669-f011:**
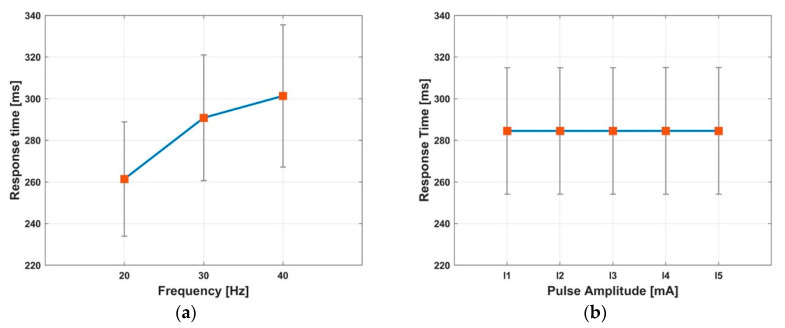
The response time (average ± standard deviation) with respect to frequency (**a**) and pulse amplitude (**b**).

**Table 1 sensors-20-05669-t001:** EMS parameter values used in the experiment.

Type	Parameter	Value
Static	Waveform	Biphasic rectangular
	Pulse Width (pd)	400 μs
Control	Frequency	20, 30, 40 Hz
	Pulse Amplitude	[$I_{1}$ ~ $I_{5}$] ^1^

^1^ Since each subject has a different value, we found the range [I1, I5] before the experiment.

**Table 2 sensors-20-05669-t002:** The peak force values (N) extracted from the measured force profile in the experiment. The values in parenthesis represent the corresponding amplitude (mA) of the electrical stimulation.

Subject	20 Hz		30 Hz		40 Hz	
	Min	Max	Min	Max	Min	Max
K01	1.47(8) *	8.95(12)	1.68(8)	12.8(12)	2.16(8)	15.6(12)
K02	1.69(12)	5.31(18)	2.10(12)	6.61(18)	2.48(12)	7.66(18)
K03	8.57(10)	23.7(15)	10.8(10)	26.7(15)	11.0(10)	28.0(15)
K04	3.77(9)	16.9(13.5)	5.21(9)	21.7(13.5)	5.82(9)	25.3(13.5)
K05	4.45(10)	13.6(15)	4.77(10)	15.3(15)	5.13(10)	15.6(15)
K06	2.80(8)	14.9(12)	4.13(8)	18.3(12)	4.69(8)	19.9(12)
K07	2.45(10)	7.67(15)	2.80(10)	11.6(15)	2.94(10)	11.7(15)
K08	3.69(8)	13.5(12)	5.81(8)	16.1(12)	6.93(8)	17.0(12)
K09	4.03(7)	8.69(10.5)	5.79(7)	11.8(10.5)	5.42(7)	12.8(10.5)
K10	2.45(7)	5.61(10.5)	3.43(7)	6.92(10.5)	3.88(7)	8.24(10.5)

* Format: Peak force (Amplitude).

**Table 3 sensors-20-05669-t003:** Parameters of exponential estimation of peak force–amplitude relation and the accuracy of the estimation at the studied frequencies averaged for all subjects.

Frequency (Hz)	a (N)	b $\left(mA^{-1}\right)$	$R^{2}$	RMSE (N)	NRMSE (%)
20	0.497(0.457) *	0.275(0.076)	0.96(0.029)	0.72(0.53)	8.35(3.37)
30	0.715(0.718)	0.278(0.099)	0.97(0.019)	0.73(0.47)	7.27(2.01)
40	0.734(0.730)	0.288(0.118)	0.97(0.025)	0.90(0.62)	7.76(3.05)

* Format: Average (Standard deviation).

**Table 4 sensors-20-05669-t004:** Estimated parameter values and accuracy of the force response model.

Subject	A (N/s)	K	$\tau_{c}\;(\mathrm{ms})$	$\tau_{1}\;(\mathrm{ms})$	$\tau_{2}\;(\mathrm{ms})$	$R^{2}$	RMSE (N)	NRMSE (%)
K01	121.9	0.300	20.8	33.1	194.4	0.97	0.64	5.97
K02	75.9	0.128	15.6	76.2	52.0	0.96	0.43	7.22
K03	438.9	0.326	31.1	40.5	51.8	0.93	2.13	9.07
K04	183.1	0.094	15.9	50.2	117.9	0.97	1.20	6.28
K05	130.8	0.012	11.5	5.0	124.1	0.89	1.47	10.3
K06	148.7	0.162	22.5	14.4	17.6	0.95	1.15	7.22
K07	85.0	0.113	17.0	11.8	189.8	0.91	0.91	9.41
K08	218.0	0.190	26.3	10.0	98.4	0.90	1.41	9.74
K09	122.2	0.215	21.9	24.1	137.4	0.89	0.93	9.30
K10	68.5	0.293	29.3	20.7	163.6	0.94	0.40	6.06
Ave. (SD)	159.3 (121)	0.180 (0.110)	21.2 (6.36)	28.6 (23.0)	130.5 (55.2)	0.93 (0.032)	1.07 (0.53)	8.06 (1.67)

**Table 5 sensors-20-05669-t005:** Result of the $R^{2}$ and NRMSE at studied amplitudes for all subjects.

Subject	$R^{2}\;(\mathrm{NRMSE})$				
	$I_{1}$	$I_{2}$	$I_{3}$	$I_{4}$	$I_{5}$
K01	0.86(14.7)	0.77(16.4)	0.76(19.8)	0.85(14.4)	0.97(5.97)
K02	0.89(10.0)	0.87(8.67)	0.91(11.9)	0.87(13.6)	0.96(7.22)
K03	0.78(17.2)	0.88(14.3)	0.91(8.99)	0.94(8.82)	0.93(9.07)
K04	0.79(13.8)	0.92(10.9)	0.89(11.0)	0.96(6.65)	0.97(6.28)
K05	0.79(12.3)	0.84(11.0)	0.80(16.0)	0.90(11.9)	0.89(10.3)
K06	0.77(14.0)	0.95(7.22)	0.86(16.2)	0.82(18.4)	0.95(7.22)
K07	0.83(13.5)	0.78(16.6)	0.81(15.3)	0.85(13.8)	0.91(9.41)
K08	0.72(11.9)	0.94(8.32)	0.88(12.2)	0.82(14.1)	0.90(9.74)
K09	0.89(11.5)	0.91(8.51)	0.85(11.9)	0.85(11.4)	0.89(9.30)
K10	0.92(6.92)	0.94(6.32)	0.91(7.44)	0.92(7.44)	0.94(6.06)
Average	0.82(12.6)	0.88(10.8)	0.86(13.1)	0.88(12.0)	0.93(8.06)

**Table 6 sensors-20-05669-t006:** Response time (±SD) of the force response for all subjects.

Subject	$T_{0.1}$	$T_{0.5}$	$T_{0.9}$	$T_{0.5}-T_{0.1}$	$T_{0.9}-T_{0.5}$	$T_{RT}$
K01	15.3(4.15) *	113.0(9.78)	364.3(38.3)	97.6(5.77)	251.3(29.1)	349.0(34.2)
K02	10.6(1.72)	77.3(6.12)	261.5(4.16)	66.7(4.41)	194.9(0.87)	250.9(4.86)
K03	8.67(1.19)	61.5(2.43)	193.0(13.8)	52.8(1.32)	131.5(11.4)	184.3(12.7)
K04	13.6(2.32)	97.4(10.0)	335.6(16.9)	83.8(7.78)	238.2(7.39)	322.0(14.6)
K05	12.0(1.34)	80.8(6.39)	277.3(8.87)	68.8(5.05)	196.5(2.75)	265.3(7.54)
K06	15.3(2.75)	108.0(5.58)	347.7(26.5)	92.6(2.89)	239.7(20.9)	332.4(23.8)
K07	15.1(3.34)	106.2(13.4)	359.7(32.5)	91.1(10.1)	253.5(19.1)	344.6(29.2)
K08	8.87(1.46)	64.1(4.16)	195.3(17.1)	55.2(2.79)	131.3(13.0)	186.4(15.7)
K09	12.3(2.47)	85.3(10.7)	292.4(20.9)	73(8.25)	207.1(10.6)	280.1(18.5)
K10	15.2(2.87)	102.6(7.90)	334.7(18.6)	87.4(5.06)	232.1(10.8)	319.5(15.8)
Ave. (±SD)	12.7 (2.63)	98.6 (37.6)	296.2 (63.9)	71.7 (27.9)	207.6 (45.3)	283.5 (61.3)

* Unit: millisecond.

**Table 7 sensors-20-05669-t007:** Response time according to the frequency for all subjects.

Subject	$T_{RT}\;\left[\mathrm{ms}\right]$		
	20 Hz	30 Hz	40 Hz
K01	304.8 *	357.8	384.2
K02	255.4	262.4	266.8
K03	167.0	193.6	192.4
K04	303.2	325.6	337.2
K05	255.6	267.0	273.2
K06	303.6	333.6	359.8
K07	305.4	357.8	370.6
K08	165.0	196.0	198.4
K09	254.8	291.0	294.4
K10	299.4	322.9	336.28
Ave. (±SD)	261.4 (54.9)	290.8 (60.4)	301.3 (68.4)

* Unit: millisecond.

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
