# Peer review of "Electrically Elicited Force Response Characteristics of Forearm Extensor Muscles for Electrical Muscle Stimulation-Based Haptic Rendering"

_sensors, 2020, doi:10.3390/s20195669_

Round 1

Reviewer 1 Report

Broad comments

The manuscript proposes a simple model (developed by others in the field of rehabilitation) to predict the actual hand extension force in response to electrical stimulation of some forearm muscles. The model was tested against some experimental measurements. The model could support haptic application (e.g. virtual reality).

The topic is interesting and has potential practical application. The manuscript is well written and clear. Only some issues need to be addressed in more details.

In particular, the electrical stimulation reported in the is study supposedly stimulates not a single muscle but rather muscle groups. Furthermore, as the stimulation parameters vary (in particular the amplitude of the impulses), more muscles and fibers are gradually involved by the electrical stimulation. The size of the electrodes and their accurate positioning is very important to predict the muscles involved and therefore the tactile sensations caused.

Instead of estimating just a rough parameter of motion (wrist extension), there are methods to accurately monitor the level of contraction of a single or more muscles. In particular, very small and thin sensors can be used that are able to measure the actual mechanical contraction (both spontaneous and that generated by an electrical stimulation). Suggested  references:

- Meglič A, Uršič M, Škorjanc A, Đorđević S, Belušič G. The Piezo-resistive MC Sensor is a Fast and Accurate Sensor for the Measurement of Mechanical Muscle Activity. Sensors 2019, 19, 2108. Doi: 10.3390/s19092108

- Esposito D, Andreozzi E, Fratini A, Gargiulo GD, Savino S, Niola V, Bifulco P. A Piezoresistive Sensor to Measure Muscle Contraction and Mechanomyography. Sensors (Basel). 2018;18(8):2553. Published 2018 Aug 4. doi:10.3390/s18082553

- Đorđević S, Tomažič S, Narici M, Pišot R, Meglič A. In-Vivo Measurement of Muscle Tension: Dynamic Properties of the MC Sensor during Isometric Muscle Contraction. Sensors 2014, 14, 17848-17863 . doi: 10.3390/s140917848

- Esposito D, Andreozzi E, Gargiulo GD, Fratini A, D’Addio G, Naik GR, Bifulco P. A Piezoresistive Array Armband With Reduced Number of Sensors for Hand Gesture Recognition. Front Neurorobot. 2020;13:114. Published 2020 Jan 17. doi:10.3389/fnbot.2019.00114

- Xiao ZG, Menon C. A Review of Force Myography Research and Development. Sensors (Basel). 2019;19(20):4557. Published 2019 Oct 20. doi:10.3390/s19204557

These considerations should be added to the manuscript (either in the introduction or even in discussion).

Specific comments

Lines 2-3 (Title): It would be better to specify that more muscle are involved in the stimulation and use the plural (i.e. Forearm Extensor Muscles).

Figure 3: please correct the words "acceleration" and "deceleration" appearing in the figure.

Lines 271-272: "Two EMS electrodes (5×5 cm, ValuTrode, Denmark) were attached to both sides of the muscle’s motor point". What Authors meant for “muscle’s motor point”? The positioning of the electrodes is not a negligible detail. It is necessary to accurately specify the position of each electrode in relation to anatomical landmarks.

Lines 273-274: "The subjects’ arm and wrist were tightly fixed on the jig with straps to constrain the motion of the forearm and …" . Did the straps hinder the stimulated muscles expansion?

Lines 276-280 and 284-285: the authors assumed that "the force measured by the sensor is equivalent to the contraction force of the extensor digitorum muscle". Actually, there are other more superficial extensor muscles of the forearm, involved in the extension of the wrist (e.g. Extensor carpi radialis longus, Extensor carpi ulnaris, etc.). Therefore, how it can be assumed with certainty that the measured force is only related to the extensor digitorum ? Please, clarify.

Line 286: How long was the time window of the moving average filter? What was filter cut-off frequency? Please, specify.

Line 308: "devise simulates" should be "device stimulates"

Line 309-310: How was the wrist flexion "detected"? Have Authors defined a sort of minimum threshold for the wrist force detected by the torque sensor, above which Authors assumed that the wrist actually performed the flexion?

Line 311: Why have Authors determined the amplitude range by fixing the frequency to 30 Hz, if this parameter actually influences the force of the elicited contraction? E.g. Authors could have found that at 20 Hz or 40 Hz the motor threshold I_mt was lower (for the same subject).

Line 315: "devise" should be "device".

Lines 337-338: what fixed time step was actually used for the numerical integration? This information could be of help for reproducibility.

Lines 350-352: from the description it seems that all samples of the force responses acquired at all three different stimulation frequencies were all put together in the parameter estimation process to extract only one couple of (a,b) values. This would be a serious flaw of the parameter estimation. However, from Table 3 in the “Results” section, it is clear that the parameter estimation has been performed separately for the peak forces obtained with the three different stimulation frequencies. Therefore, I suggest clarifying in lines 350-352 that the estimation of the (a,b) parameters values has been performed separately for the peak forces obtained in response to the three different stimulation frequencies.

Lines 408-412: Do the measured and predicted peak forces include the results obtained in response to all the three different stimulation frequencies at a determined amplitude (e.g. does Figure 7.(a) include all the peak forces obtained in response to pulse trains at 20/30/40 Hz with amplitude equal to I1)? Why was it necessary to estimate the correlation between measured and predicted peak forces separately for each stimulation amplitude, instead of considering all the measured peak forces versus all the corresponding predicted values at all combinations of pulse amplitudes and frequencies?

 Lines 412-414: Similarly to the previous comment, do the measured and predicted peak forces include the results obtained in response to all the five different stimulation amplitudes at a determined frequency? Again, why was it necessary to estimate the correlation between measured and predicted peak forces separately for each stimulation frequency, instead of considering all the measured peak forces versus all the corresponding predicted values at all combinations of pulse amplitudes and frequencies?

Line 436: Please, check the caption of Table 4

Figure 9: The estimated force responses clearly show oscillations at the particular stimulation frequency, which decrease in amplitude with increasing frequency. The corresponding experimental force responses do not show such oscillations at all. This can be observed also in Figure 10. Please, add some comments about the issue (e.g. the model is simplified)

Lines 441-446: Why didn’t Authors estimate the parameters of the force response model also at the other amplitudes and frequencies, so as to compare them with the results obtained with parameter estimation performed at I5 amplitude and 20 Hz frequency?

Figure 10: Although the adopted lines colors are the same of Figure 9, it would be better to include a legend that clearly indicates which lines correspond to experimental and predicted results, as done in Figure 9. It is advisable to mention the colors used also in the caption of the images.

Lines 457-460: Where was the time t = 0 set, in order to determine the times T0.1 and T0.9? This have to be clearly explained in the manuscript.

Lines 460-461: Was the response time averaged on all the values estimated from the force responses corresponding to all combinations of pulse amplitudes and frequencies? Were all other times (T0.1, T0.5 T0.9) obtained with the same procedure? This should be clarified. However, it would be better to report also the standard deviation.

Figure 12: It should be referred to as "Figure 11", as already done in the text at line 467.

Lines 494-495: The reported results do not correspond to those presented in Table 4

Line 500: According to the results, "identical" is too strong of a word: I suggest replacing with a more soft expression.

Lines 527-528: Which of the presented results does support this statement? T0.1, which, in principle, could be considered as an "onset time" of the muscle contraction with respect to the electrical stimulus, turned out to be about 13 ms on average, as reported in Table 6.

Lines 534-535: This statement is not very clear. Please clarify what Authors meant to say here. What do Authors refer to with "time lag"? Which events do actually define this lag?

Lines 536-538: Although they could be derived from the results reported in Table 6, the quantities "T0.9-T0.5" and "T0.5-T0.1" were not explicitly reported in Table 6, so it is not straightforward for the reader to verify the statement at lines 536-538.

Reviewer 2 Report

In this manuscript, electrically elicited force response characteristics of forearm extensor muscle for electrical muscle stimulation-based haptic rendering were studied. The study is interest to readers. The experimental results are good. This manuscript is acceptable for publish.

Author Response

Thank you for reviewing the manuscript.

Reviewer 3 Report

This paper is very well written with nicely described all manuscript parts including limitation part. The figures have good visibility as well and are understandable.

It describes analyzed force response of forearm extensor muscles for EMS-based haptic rendering by introducing a simplified force response model with amplitude modulation using an exponential function.

The analysis of the force response characteristics and the application of this model can may help enhance the fidelity of EMS-based haptic rendering.

Author Response

Thank you for reviewing the manuscript.

Reviewer 4 Report

The topic is timely and a lot of questions remain in this research area. Nevertheless, I have the following comments and concerns:

Introduction

  • From line 42: Authors should strongly reduce this part and be focused on the relevant mechanical configuration that is related to the topic of the article. It should be easier for readers. Instead of that, the relevance of using EMS as a haptic device should be highlighted. Likewise, the “force response model” is complicate to understand in Introduction. What is its relevance? Which applications? It is an issue for what?
  • Line 67: « Artificial muscle contraction due to electrical stimulation can lead to rapid muscle fatigue compared with voluntary contraction ». What do you mean by « artificial muscle contraction »? EMS is characterized by a significant physiological response including muscle damage and then a muscle reinforcement. It also induces neuromuscular adaptations (in spinal cord and brain) responsible of muscle force production. Is it really artificial?
  • Line 71: « …limiting the degree of stimulation to certain thresholds [26]… ». What did the authors mean by « certain thresholds ». Please be more explicit.
  • Line 93: “In the field of rehabilitation, various mathematical models representing the relationship between the electrical stimulation and the resulting force of muscle contraction have been proposed”. It is hard to understand what is the relevance of the mathematical models in the field of rehabilitation. Please be more explicit. The Introduction needs to be re-written.
  • The main thread and the purpose need to be clearly written.

Methods

  • Line 125: This part is too long. Please focus on the main principles.
  • Accerlation should be defined (typing error?)
  • Experimental Procedures are not clear. Please explain what is done during experimentation in humans.
  • How many subjects are included? Healthy subjects, age, sex?

Results and discussion

  • These parts are hard to read and should be clarify by staying focused on the main points.
  • Methods should not be repeated in result section.
  • The limitation section is once again not enough readable.
  • There are a lot of repetitions through the manuscript until the conclusion. Please be more concise by first avoiding repetitions and then by being more explicit.
  • The clinical applications are not clear and the demonstration of this haptic device remains confused until the end of the conclusion.

Round 2

Reviewer 4 Report

Regarding the manuscript « sensors-937442 », the authors have appropriately addressed my concerns from initial review.

I'm not a native english speaker but this manuscript needs to be read again by a native english speaker to facilitate the reading.